# Optimization of Enterovirus-like Particle Production and Purification Using Design of Experiments

**DOI:** 10.3390/pathogens14020118

**Published:** 2025-01-27

**Authors:** Louis Kuijpers, Wouter J. P. van den Braak, Abbas Freydoonian, Nynke H. Dekker, Leo A. van der Pol

**Affiliations:** 1Department of Bionanoscience, Delft University of Technology, Van der Maasweg 9, 2629 HZ Delft, The Netherlands; 2Intravacc B.V., Antonie van Leeuwenhoeklaan 9, 3721 MA Bilthoven, The Netherlands; 3Department of Physics and Kavli Institute of Nanoscience Discovery, University of Oxford, Oxford OX1 3QU, UK

**Keywords:** coxsackievirus A6, enterovirus A71, optimization, design of experiments, hand, foot, and mouth disease

## Abstract

Hand, foot, and mouth disease (HFMD) represents an emerging health concern whose main causative agents are Coxsackievirus A6 (CVA6) and enterovirus A71 (EV71). The lack of a CVA6 vaccine and the rise of new HFMD-causing strains due to the containment of established HFMD-causing viruses necessitates the search for alternative vaccine technologies, including virus-like particle (VLP) vaccine candidates. While studies have demonstrated that production of enterovirus-like particles in various organisms can be achieved by expression of the viral P1 structural proteins and the 3CD protease, optimization based on the interplay between the three most commonly altered infection parameters (multiplicity of infection (MOI), viable cell density at the time of infection (VCD), and the infection period) is often not investigated. To address this challenge, we have performed Design of Experiments (DoE) to optimize the production of CVA6 and EV71 VLPs. Our results indicate that CVA6 VLP production peaks at high MOI, high VCD, and long infection periods. Our subsequent downstream purification processes yielded 38 mg and 158 mg of purified CVA6 and EV71 VLPs from 1 L crude harvest, respectively. This translates into thousands of potential vaccine doses and highlights the economic potential of enterovirus-like particles for vaccine purposes.

## 1. Introduction

Over the past two decades, hand, foot, and mouth disease (HFMD) has emerged as a growing health concern, particularly in South-East Asia. Multiple outbreaks in Taiwan, Malaysia, China, Vietnam, and Cambodia have demonstrated the emergence and spread of HFMD, alarming medical experts [1]. The disease is named after the characteristic lesions that develop on the hands, feet, mouth, and buttocks of patients following infection. In some cases, especially in children, the disease can spread to the central nervous system (CNS), leading to severe complications such as aseptic meningitis and encephalitis [1,2]. Coxsackievirus A6 (CVA6) and enterovirus A71 (A71) are the main causative agents of HFMD. For CVA6, the most prevalent strain detected in approximately 50% of HFMD cases, no vaccine is available at present [3,4,5]. For EV71, substantial progress has been made through the development and employment of an inactivated EV71 vaccine, which effectively suppresses the virus post-infection and curbs viral outbreaks [6]. However, while this vaccine has multiple advantages compared to attenuated virus vaccines containing live viruses, it also presents some distinct drawbacks. Firstly, the long virus inactivation procedure required (on average, two weeks for enteroviruses) renders the production process and the realization of the end product highly costly [7]. Secondly, suppressing only one virulent strain provides the opportunity for other virus strains to emerge, considerably reducing the efficacy of the vaccine in the medium term [4,5]. This continuous cycle can only be interrupted by multivalent vaccines that target multiple viral strains simultaneously and eradicate the disease.

Various vaccine development approaches have been proposed to obtain multivalent vaccines against HFMD, with a recent strategy involving the utilization of virus-like particles (VLPs). Unlike conventional inactivated or attenuated vaccines, VLPs are macromolecular (self-)assemblies of viral structural proteins that do not contain the viral genome, rendering them non-infectious and incapable of replication. Their surface structure resembles the immune-inducing patterns of the native virus and thus enables recognition by the immune system [8,9]. This characteristic can be leveraged for vaccine purposes, particularly for viruses that are identified by the immune system based on specific surface patterns (e.g., enteroviruses) as opposed to a single protein [10,11].

The genomes of EV71 and CVA6 consist of a ~7.4 kbp positive sense ssRNA, which encodes 11 viral proteins (Figure 1A). These proteins are expressed as a 260 kDa polyprotein that contains three regions: P1, P2, and P3. The P2 and P3 regions are composed of non-structural proteins with mixed functionalities, ranging from protease to polymerase activity. The viral capsid is composed of 60 structural assemblies of proteins that originate from the P1 region [12,13]. During morphogenesis, the P1 is proteolytically cleaved in a stepwise fashion into the VP0 (the precursor of VP2 and VP4), VP1, and VP3. Subsequently, VP0 is autocatalytically cleaved into VP2 and VP4 to complete the viral maturation process. VP1 is typically used as a benchmark protein to quantify enterovirus viral protein and virion/VLP production, and we follow this approach in this work. Furthermore, given the essentially identical atomic structure between the procapsid (virion precursor) and the A-particle (uncoating intermediate) and the observation that VLPs closely resemble procapsids (both hypothesized to lack RNA), the A-particle provides a second benchmark for VLP production [12].

Here, we describe a systematic approach to optimize virus-like particle (VLP) yield using the baculovirus expression vector system (BEVS). Previous studies have shown that the expression of the P1 structural proteins and the 3CD protease is sufficient to produce enterovirus-like particles in various organisms (Figure 1A) [14,15,16,17,18,19]. While construct optimization has been reported for the production of enterovirus VLPs, optimization based on culturing and infection conditions is mostly limited to the multiplicity of infection (MOI) and infection period (t_inf_) [16,20,21,22]. The viable cell density at the time of infection (VCD) is usually not considered, and the interplay between these three infection parameters is also rarely evaluated. To address this problem, we performed Design of Experiments (DoE) to optimize the production of CVA6 VLPs. A DoE is a systematic, cost-efficient, and effective approach that allows researchers to investigate the relationship between multiple input factors (e.g., MOI, VCD, and t_inf_) and responses (e.g., viral protein yield). It is an iterative process in which center points and ranges for relevant independent process parameters are varied to deduce a more favorable set of input factors (Figure 1B). The DoE not only provides insight into the optimized conditions but additionally presents a model for the estimation of viral protein yields at conditions (within the boundary conditions of the DoE) that are not experimentally tested. The parallel nature of the approach significantly reduces the process optimization time compared to performing successive experiments [23]. Additionally, biological variation originating from storage and cell conditions is minimized. Various models can be employed for the DoE, but for initial discovery and optimization, the full factorial model is the gold standard (Figure 1B).

## 2. Materials and Methods

### 2.1. Construction of Recombinant Baculoviruses

Previously reported construct optimizations for the production of enterovirus A71 (EV71) indicated that the highest viral protein yields were obtained using the P1 and 3CD coding sequences behind the polh and CMV promoters, respectively [21]. Two transfer plasmids containing the aforementioned genes originating from EV71_C4 and CVA6, respectively, were amplified using *E. coli* JM109 competent cells (Promega, Leiden, The Netherlands) and isolated using NucleoBond Xtra Midi EF (Macherey-Nagel, Dueren, Germany). The amplified transfer plasmids were verified by double digestion using multiple restriction enzyme pairs. To construct the recombinant baculovirus, the transfer plasmid and *flash*BAC GOLD DNA (Oxford Expression Technologies, Oxford, UK) were co-transfected into Sf9 cells (Gibco, Paisley, Scotland). The two different baculovirus stocks for the EV71 and CVA6 origin viral proteins were designated BacV-EV71 and BacV-CVA6, respectively. The baculovirus was propagated over the course of three passages and titered by the end-point-dilution method using Sf9 Easy Titer (Sf9ET) (Kerafast, Shirley, MA, USA) cells according to the manufacturer’s protocol.

### 2.2. Viral Protein Production & Purification

EV71 and CVA6 VLP production was accomplished by infection of High Five^TM^ cells (Gibco). Erlenmeyer culture flasks (125 mL) with 50 mL working volumes of Hi5 cells (cell density-dependent on the specific experiment) were infected by either BacV-EV71 or BacV-CVA6 (MOI dependent on the experiment) in Sf-900IISFM (Gibco). After the infection period (dependent on the experiment), the cultures were harvested using centrifugation (Heraeus, Hanau, Germany) at 3000 rpm for 15 min at 4 °C. The pellet was discarded, and the supernatant was supplemented with Triton X-100 (final concentration 0.1%) to eliminate cell-derived external vesicles and destabilize and inactivate baculoviruses. The viral protein production optimization analysis was performed on this sample. For purification optimization, the most successful protocol started from this point.

The Triton X-100 supplemented supernatant (2–5 mL) was filtered using 0.45 µm and 0.22 µm filters (Merck, Darmstadt, Germany). The filtered supernatant was subjected to sucrose cushion ultracentrifugation (30% *W/W* sucrose in PBS) for 5 h at 141,000× g, and 4 °C. UC was performed using Beckman Coulter centrifuge tubes (Cat#: 355642) in an SW28 Beckman Coulter rotor. The pellet was resuspended in 1 mL phosphate-buffered saline (PBS, Gibco) and subjected to another centrifugation (10,000× g, 10 min), after which the pellet was discarded, and the supernatant (1–2 mL) was applied onto a discontinuous sucrose density gradient (SDG) (15–45% *W/W* sucrose in PBS) and ultracentrifuged for 3 h at 141,000× g and 4 °C (ThermoFisher Scientific, Bleiswijk, The Netherlands). In addition, UC was performed using Beckman Coulter centrifuge tubes (Cat#: 355642) in an SW28 Beckman Coulter rotor (Woerden, The Netherlands). The SDG was harvested by 1 mL sampling from the top. Samples were pooled into eight concentration ranges based on the percentage of sucrose (0–15%, 15–20%, 20–25%, 25–30%, 30–35%, 35–40%, 40–45%, and 45–50%), whereby sucrose density was measured by refractometer (Hanna instruments, Bedfordshire, UK).

### 2.3. SDS PAGE & Western Blot

To confirm the viral protein production of EV71 and CVA6, VP1 (both origins) Western blot analysis and SDS PAGE were performed. For this, 40 µL of each culture was mixed with 10 µL loading buffer (Intravacc, Bilthoven, The Netherlands) and heated at 100 °C for 15 min. Then, 15 µL of each mixture was loaded on NuPAGE^TM^ 10% Bis-Tris gel (Thermo Fisher Scientific, Bleiswijk, The Netherlands), and electrophoresis was performed at 200 V for 45 min. The gel was washed three times for 5 min using purified water. Depending on the purpose of the gel, it was subjected to either staining or overnight blocking (see sections below).

#### 2.3.1. SDS PAGE

The gel was stained with Imperial^TM^ protein stain (Thermo Fisher Scientific) for 1 h and destained in multiple cycles and overnight using purified water.

#### 2.3.2. Western Blot

The proteins were blotted onto a nitrocellulose membrane (Thermo Fisher Scientific) using a semi-dry blotting machine (Hoefer, Bridgewater, MA, USA) run at 60 mA for 60 min. The membrane was washed two times in wash buffer (PBS + 0.1% *W*/*W* Tween 20 (Sigma-Aldrich, Zwijndracht, The Netherlands)) and blocked overnight at 4 °C (PBS + 0.1% Tween 20 (Sigma-Aldrich), and 0.5% Protifar (Nutricia, Zoetermeer, The Netherlands)). Next, the membrane was incubated with 1:1000 of either mouse anti-EV71-VP1 (Abnova, Cambridge, UK, Cat# MAB1255-M08) or rabbit anti-CVA6-VP1 (Thermo Fisher Scientific, PA5-112001) in block buffer for 90 min on a roller incubator. The membrane was washed five times using wash buffer and incubated with 1:2000 of either goat anti-mouse IgG Human ads-AP (Southern Biotech, Birmingham, AL, USA, Cat# 1030-04) or goat anti-rabbit IgG Human ads-AP (Southern Biotech, Cat# 4050-04) in block buffer for 90 min on a roller incubator. The membrane was washed three times using wash buffer and once using purified water. The membrane was colored using an AP conjugate substrate kit (BIO-RAD, Lunteren, The Netherlands, Cat# 1706432). The colorization reaction was terminated by the addition of an excess of purified water.

### 2.4. Enzyme-Linked Immunoassay

EV71 VP1 ELISA was performed using an ELISA kit (Abnova, Cambridge, UK, Cat# KA1677) according to the manufacturer’s instructions.

To confirm the Western blot results for CVA6, an ELISA using anti-CVA6 A-Particle (Creative Biolab, Frankfurt am Main, Germany) was performed. To this end, 100 µL of the sample was added to a 96-well high-binding ELISA plate (Greiner Bio-One, Alphen aan de Rijn, The Netherlands) and 1:1 serial diluted in PBS. The plate was incubated at RT overnight. The plate was subjected to three wash cycles (purified water + 0.05% Tween 80 (Sigma-Aldrich)). Plates were blocked using 150 µL of block buffer (PBS + 0.5% bovine serum albumin (BSA, ThermoFisher Scientific, Bleiswijk, The Netherlands)) for 30 min at 37 °C. Each plate was then subjected to three wash cycles using a washer dispenser (Agilent, Santa Clara, CA, USA, Cat# EL406). The primary antibody (recombinant mouse anti-CVA6 A-particle, Creative Biolabs, Frankfurt am Main, Germany, Cat# PABC-483) was added in a final concentration of 0.1 µg/mL in 100 µL/well assay buffer (PBS + 0.05% Tween 80, ThermoFisher Scientific, Bleiswijk, The Netherlands) and incubated at 37 °C for 1 h. The plates were washed three times, after which the secondary antibody (goat anti-mouse IgG-HPR (Southern Biotech, 1030-05) was diluted 5000-fold in PBS, and 100 µL of the result per well was added to the plate and incubated at 37 °C for 1 h. The wells were washed three times, and 100 µL of HRP substrate (Ultra TMB-ELISA (Thermo Fisher Scientific)) was added to each well and incubated for 10 min in RT. To quench the reaction, 100 µL of 0.2 M sulfuric acid was added to each well. The wells’ light absorption was measured at 450 nm using a plate reader (synergy neo2 reader, Agilent, Santa Clara, CA, USA).

### 2.5. Design of Experiments

To investigate the optimized conditions and analyze the output for viral protein production in terms of MOI, VCD, and infection period (the three DoE input factors), a Design of the Experiments was performed using the full-factorial model of model software MODDE12 (Sartorius).

### 2.6. Electron Microscopy

Negative stain transmission electron microscopy was employed for the direct visualization of VLPs. Of each DSP fraction, 3 µL sample solution was dispensed on glow-discharged EM grids (Quantifoil, carbon-supported, Cu-400, Großlöbichau, Germany), incubated for 60 s, and blotted away using filter paper. This procedure was repeated three times to increase the particle concentration on the carbon support. Next, grids were subjected to two wash cycles using 10 µL PBS. Excess liquid was blotted away using filter papers. Staining was performed using 2% uranyl acetate for 1 min. Excess liquid was blotted away using filter paper. Imaging was performed using a JEOL JEM-1400plus TEM (Tokyo, Japan) operated at 120 kV, and micrographs were acquired on a TVIPS F416 CMOS camera (Karlsruhe, Germany) at multiple magnifications.

### 2.7. Bicinchoninic Acid (BCA) Total Protein Assay

Bovine Serum Albumin (BSA), with a stock concentration of 2 mg/mL, was subjected to serial dilution in PBS to generate a standard curve ranging from 0 to 320 µg/mL. Samples were appropriately diluted in PBS to ensure that their concentrations fell within the linear range of the standard curve. Subsequently, 25 µL of each sample or standard was added to a 96-well plate, followed by the addition of 200 µL of BCA reagent mixture (a combination of BCA reagent A and B in a 40:1 ratio, sourced from the Pierce™ BCA Protein Assay Kit, Thermo Fisher Scientific) to each well. The plate was then incubated at 37 °C for 30 min, after which the absorbance was measured at 562 nm, and the resulting values were used to calculate the protein concentration using the standard curve.

## 3. Results

### 3.1. Primary Iteration of DoE for CVA6

Design of experiments (DoE) is a powerful tool for systematically optimizing desired outputs (or responses), such as maximizing viral protein yield, minimizing protein aggregation, minimizing denaturation, or cell death, based on specifically chosen process parameters (or factors). Here, the chosen (input) factors were a multiplicity of infection (MOI), viable cell density at the time of infection (VCD), and infection period (t_inf_). In the first iteration of experiments, the boundary conditions for each factor were set based on extremes found in the literature [14,15,16,17,18,19,20,22,24]. In addition, a mock infection, serving as a negative control, and three triplicate experiments of the center points of the selected condition were included. These triplicates were designed to assess the level of biological variation between the experiments. The experimental conditions are summarized in Table 1. Since DoE optimization is required for each viral strain independently and given that many parameters have already been investigated for EV71, in this study, we focused exclusively on the outcomes of the DoE for CVA6.

The DoE was performed to optimize extracellular viral protein (i.e., VLP) production, which yields less complex downstream processing (DSP) than intracellular VLPs. Additionally, significant efforts are being made to enhance the secretion of expression products, thereby rendering the investigation of extracellular production a more compelling approach compared to intracellular production [25,26]. The extracellular viral protein production was read out using western blot (Figure 2A). Western blots indicated the presence of VP1 at the anticipated molecular weight of 36 kDa, as reported in the literature [17,27]. Although numerous *Picornaviridae* share the same 11 proteins (Figure 1A), slight variations in size for the individual proteins among viruses within a genus are frequently observed [27,28]. Different concentrations of VP1 were measured under different experimental conditions. Notably, no VP1 was observed in the mock infections (negative controls, Δvirus).

To quantitatively determine which of the cultures produced the highest amount of VP1, band intensities were measured using GelAnalyzer software [29] and normalized to the highest output (Figure 2C). This allowed for comparison between the different cultures. The triplicates (experiments 9–11) indicated highly similar values, suggesting a relatively low level of biological variance. The experiment with the highest intensity band was found to be experiment 8. The corresponding input factors were on the high end of the boundary conditions. Furthermore, other infection experiments with long infection periods (exp. 3 and 4, and to a lesser extent, the triplicates) also indicated high concentrations of VP1 for CVA6, indicating a preference for extended infection periods independent of MOI or VCD.

An ELISA was performed to independently verify the western blot results (Figure 2A). It needs to be noted that the CVA6 A-particle ELISA probes for the presence of assembled particles. Nevertheless, the ELISA showed a similar trend in the normalized values as the western blot (Figure 2C, dark blue). Moreover, this suggests that for CVA6, regardless of the infection conditions, the same ratio existed between the total produced VP1 protein and assembled VLPs (Figure 2C, experiments 3, 4, and 8–11). We note that minor signals detected in the western blots were not detected by ELISA (Figure 2C, experiments 1, 2, and 5–7); this could be explained by non-VP1-specific bands being quantified as VP1. Additionally, the differences in the CVA6 ELISA in comparison to the CVA6 western blot (Figure 2C) were attributed to the target difference of the antibodies (VP1: Figure 2C light blue versus A-particle: Figure 2C dark blue). Two experiments for CVA6 (Figure 2C, experiments 2 and 6) were found to deviate from the established pattern, as they show high western blot scores but no corresponding ELISA signal. Apart from the non-specific band quantification described above, these results could be attributed to the high MOI and short infection period used, which may have favored viral protein production but hindered VLP assembly. Either the time for assembly or the release of VLPs into the extracellular environment was insufficient. Overall, the data once again highlighted the importance of extended infection periods for efficient VLP production.

From the first iteration of DoE, it could be concluded that the optimal conditions for CVA6 VLP production were a long infection period, high MOI, and high cell density. A second iteration of DoE was conducted to further optimize viral protein expression using the BEVS (Figure 1B). In this second iteration, the lower boundaries of the input factors MOI and infection period were increased, whereas the value of the VCD input factor was decreased. The high VCD could have been attributed to the large amount of contaminating cell debris observed during harvesting, making it significantly harder to purify the protein of interest. To facilitate future work, we lowered the VCD to counteract these effects.

### 3.2. Secondary Iteration of DoE

The second iteration of the viral protein production optimization for CVA6 was performed based on the experimental conditions outlined in Table 2. Western blot analysis was performed on samples from each extracellular environment, and the results were quantified using GelAnalyzer software, as shown Figure 2D. In this iteration, all samples except the negative control (experiment 12) indicated VP1 presence (Figure 2D). The highest yields were obtained using high MOI (5), VCD (1 × 10^6^ cells/mL), and infection period (7 d).

As stated previously, apart from determining the best infection conditions, the MODDE [30] analysis software also provides a model for the viral protein yield in experimental conditions that were not physically carried out. The performance of this model was assessed based on four key parameters, which were summarized in the summary of fit plot (Figure 3A):R^2^ = 0.98, a value greater than 0.5, showing statistical significance.Q^2^ = 0.77, a value greater than 0.5, indicating a good model. Additionally, the difference between R^2^ and Q^2^ was not larger than 0.3, further indicating a good model.Model validity (MV) = 0.65, indicating an absence in Lack of Fit, outliers, and transformation problems.Reproducibility = 0.97, indicating minimal biological variance.Overall, it could be concluded that the model developed for CVA6 was able to accurately predict viral protein production based on the input factors.

The CVA6 model demonstrated satisfactory performance, as shown in the model performance indicators (Figure 3A). Subsequently, the contour plots for viral protein yield at three different measured MOIs were generated for CVA6 (Figure 3B), which predicted the viral protein production based on the model and allowed for the estimation of yields under unmeasured conditions. The plot also confirmed the western blot results and indicated that higher yields could be obtained with longer infection periods and higher VCD. Increases in these input factors might further optimize VLP production but could also result in an increase in cell debris that complicates the purification process.

### 3.3. Optimizing the Purification Process

Following successful viral protein production using these optimized conditions, an optimization study was conducted to improve the purification process. High purity of the VLPs is critical to ensure a high quality product and enable visualization using electron microscopy (EM). Here, we describe multiple strategies for the downstream processing (DSP) of EV71 and CVA6 VLPs. The DSP of VLPs involves multiple steps, including clarification, sucrose cushions/density gradients, and centrifugation. Figure 4 presents the stepwise optimization strategy for obtaining the desired purity for EV71 VLPs (alterations in the process are indicated in red). Previous work suggests that about one-third of the total viral protein produced is released into the extracellular environment, while two-thirds remain intracellular [21]. Whether VLPs are actively transported out of the cells or released due to cell lysis over time remains unknown. Designing a DSP strategy for the purification of extracellular VLPs may result in product loss but could significantly reduce the purification process time and effort.

The initial strategy for purifying extracellular VLPs, based on previous work, reported highly pure material after only a few DSP steps [16]. To eliminate extracellular vesicles that are produced by the insect cells as a contaminating byproduct [31] and to disrupt and (partially) inactivate the baculoviruses in the crude supernatant, an additional triton-treatment step was added to the published protocol (Figure 5) [32,33], yielding strategy A (Figure 4A). It is important to note that the use of the Hi5 cell line significantly reduces the production of extracellular vesicles; however, their presence remains detectable (Figure 5) [31]. Analysis of the final sucrose density gradient (SDG) fractions indicated a very low protein concentration, which was barely visible on SDS PAGE gel (Figure 6A). Fortunately, the number of sample-contaminating proteins was low. Moreover, on western blot, the VP1 band that indicated viral protein expression was visible (Figure 6A). Direct observation of the most promising fraction (25–30%) by electron microscopy indicated a relatively pure sample with the presence of VLPs (Figure 7A, red arrow and inset). Notably, while the sample did not contain any baculovirus, there was a background of contaminating structures, which originated from either the host cell or the viral proteins.

The triton treatment was moved up in the process order (strategy B, Figure 4B), to reduce sedimentation of larger structures through the 30% sucrose and contaminating the VLP-containing fractions of interest. Both the SDS PAGE gel and the western blot for this strategy looked highly similar to the results obtained when employing strategy A (Figure 6B). However, the EM images have a much clearer background (Figure 7B), with fewer aggregates. In both strategies (A and B), the particle density was low. To increase the particle density, the production cultures were scaled up from 50 mL to 400 mL, which presented new challenges for the DSP protocol.

After scaled-up viral protein production, strategy C (Figure 4C) yielded poor results. The VP1 protein could be detected in every fraction of the SDG, most likely due to VLP aggregation, binding of VLPs to host cell proteins (HCP), DNA or membrane fragments, mixing of the SDG layers during harvesting, or reduced efficiency of the SDG due to the small volume (Figure 6C). Although the particle density increased as observed by EM, so did that of the contaminants (Figure 7C). To improve separation, the volume of each sucrose fraction in the SDG was increased from 2 to 6 mL (strategy D, Figure 4D), resulting in a clear distinction between VP1-containing and VP1-lacking fractions Figure 6D). Moreover, the particle density of the fraction of interest (25–30%) increased (Figure 7D). However, aggregated structures were still present.

To eliminate these contaminants in the scaled-up viral protein production, a double filtration was introduced prior to SDG (strategy E, Figure 4E). This additional clarification step did not appear to affect the viral protein concentration, as evidenced by the high band intensity similarity between strategies D and E (Figure 6E). Additionally, the number of contaminants was significantly reduced (Figure 7E). To further enhance purity, this filtration step was shifted up in the process (strategy F, Figure 4F). However, the amount of viral protein yield at the end of the process appeared to decrease, as illustrated by the reduced band intensity on western blot (Figure 6F). Contributing to the finding that the total protein concentration was low, the SDS PAGE gel only indicated dim bands (Figure 6F). Surprisingly, EM investigation of the most promising fraction (25–30%) indicated a high particle density and purity (Figure 7F), with only marginal amounts of aggregates that could be attributed to VLP disintegration.

The results of strategy F showed the most promising outcomes for EV71 VLP purification, as evident from the low levels of contaminating proteins on SDS PAGE (Figure 6F), clear visualization of VP1 band on western blot (Figure 6F), and presence of VLPs without aggregates in the EM image (Figure 7F). The SDG fraction with the highest band intensity in strategy F (25–30%) was subjected to total protein determination analysis, and a concentration of ~15.8 mg/mL was determined (final volume ~4 mL; ~158 mg purified EV71 VLPs from 1 L crude harvest [21]).

Since the two VLPs originated from the same *Enterovirus* genus, we expected purification protocols to be universally applicable. To test this hypothesis, DSP strategy F (Figure 4F) was applied to the CVA6 VLP production batches. Unfortunately, the protocol that could successfully purify EV71 VLPs did not yield similar results for CVA6 VLPs, as demonstrated by the widespread distribution of CVA6 VP1 in the majority of the SDG fractions (Figure 8B; Strategy F; western blot). Additionally, there are contaminant proteins visible on the SDS PAGE gel (Figure 8B; Strategy F; SDS PAGE). The most plausible explanation for these observations is the presence of aggregates resulting from the high MOI, high VCD, and prolonged infection period for CVA6. To ascertain that the previously most promising fraction (25–30%) did not contain VLPs to the extent that the EV71 fractions did, EM images were taken (Figure 8C). The images revealed a clear background with large structures, effectively masking any VLP presence and supporting the hypothesis of extensive aggregation of proteins or other cellular or viral components.

To address the issues encountered when applying strategy F to CVA6 VLP purification, we designed strategy G (Figure 8A). Strategy G involved an additional filtration step prior to the sucrose cushion, which resulted in a lower concentration of viral proteins on SDS PAGE gel (Figure 8B; Strategy G; SDS PAGE). Furthermore, the gel indicated that the VP1 and VP0 bands were overlapping, VP3 was produced and presented at its correct size (~27 kDa), and analogously to all previous western blots for CVA6, a truncated version of VP1 (VP1*) was apparent [27,28]. Lastly, an additional band was visible at ~60 kDa, which could represent host cell proteins or an uncleaved VP0-VP3 protein, as previously observed for EV71 and CVA16 [24,27]. The SDS PAGE indicated high purity, as there was only a minimal number of contaminant proteins visible. The western blot showed a clear non-equilibrium distribution of VP1 in multiple fractions, with the 25–30% fractions being the most promising (Figure 8B; Strategy G; western blot). EM images indicated the presence of assembled VLPs in a large excess of the observed aggregates (Figure 8D). Overall, we concluded that strategy G yielded highly pure CVA6 VLPs in the most promising SDG fraction (25–30%). The total protein concentration of the purified CVA6 VLPs was determined to be ~3.8 mg/mL (final volume ~4 mL; ~38 mg purified CVA6 VLPs from 1 L crude harvest).

## 4. Discussion

In this study, we have presented a successful optimization of CVA6 VLP production and EV71 and CVA6 purification for potential vaccine purposes. The two optimization studies (production and purification) yielded VLPs with a high production yield and high purity.

Optimization of infection conditions was performed based on western blot band intensity and confirmed using ELISA assays. For CVA6 VLPs, two iterations of DoE were performed, and the optimum infection parameters were determined to be high MOI (5), high VCD (1.0 × 10^6^ cells/mL), and a long infection period (7 d). The infection period is relatively long in comparison to previous studies, which could contribute to more impurities in the supernatant [16,17,20,22,24,27,28,34,35]. A high MOI was observed in multiple studies, even exceeding the MOI used in this study [16,20,22,34]. Typically, a lower MOI is favored to preserve viral stocks.

We note that the model presented in this study exhibited robust performance parameters, indicating a high degree of accuracy and reliability in predicting infection outcomes. Specifically, higher VCD and longer infection periods are recommended as input factors for an additional iteration of DoE to determine an even more favorable set of infection conditions for VLP production. Additionally, a large amount of previous work has described infection with high cell density cultures (≥2,000,000 cells/mL in comparison to our 1,000,000 cells/mL for CVA6), which could potentially result in higher viral protein yields [16,17,20,27,34]. However, it is important to consider the potential drawbacks. The trade-off is that higher cell densities over long infection periods can cause cell death and result in an increased amount of cell debris, rendering VLP purification more challenging. Therefore, a careful balance between these factors must be considered in further optimizing the VLP production process. Additionally, culturing in Erlenmeyer culture flasks limits the operating mode to batch. While the medium can be replenished during the infection period, this results in a loss of product and baculovirus. Bioreactors can allow for fed-batch or even continuous feed and harvesting but could lower the viral protein yield per volumetric unit [20,24].

Optimization of the upstream process conditions and the subsequent purification process for EV71 VLPs resulted in a final yield of 158 mg purified VLP product generated from 1 L crude culture harvest. This yield represented a 2.5-fold increase over the previously reported highest yield of 60 mg purified EV71 VLPs from 1 L crude harvest [21]. With respect to the purification of our VLPs, we applied discrimination based on size and density. In the process, the impurities that seem to present the most concern are the baculoviruses and the cell-derived external vesicles (exosomes and microvesicles shed by the insect cells). The elimination of these contaminating external vesicles, a process of significant importance, as virtually every cell generates extracellular vesicles, remains underappreciated [31]. In this application of enterovirus VLP purification, it is favorable that these VLPs are non-enveloped capsid structures, which presents detergent treatment, as with Triton, as an option for destruction of vesicles and removal and inactivation of baculovirus [32,33]. A detergent treatment, as one of the earlier steps of the DSP, takes care of the removal of main impurities and provides an opportunity to remove the detergent in subsequent steps (not further shown here).

The optimization study of the purification process for CVA6 VLPs resulted in a final yield of 38 mg purified CVA6 VLPs from 1 L crude harvest, a 4-fold decrease in comparison to the EV71 VLPs due to the additional filtration step. The lack of a previously reported yield for CVA6 VLPs makes it challenging to make comparisons. A comparison with CVB1 and CVB3 VLP production, which yielded approximately 1.5–3 mg/L, indicates a superior yield achieved in this study [36,37,38].

Furthermore, our study eliminates the need for co-transfection with multiple plasmids or baculoviruses, as described in previous studies [38], simplifying the production process. Most importantly, the previously determined potent immunogenicity offered by vaccines at 10–25 µg, translates to thousands of doses per liter of culture medium, confirming the similar economic potential for CVA6 VLPs to EV71 VLPs [16,21,28]. 

## Figures and Tables

**Figure 1 pathogens-14-00118-f001:**
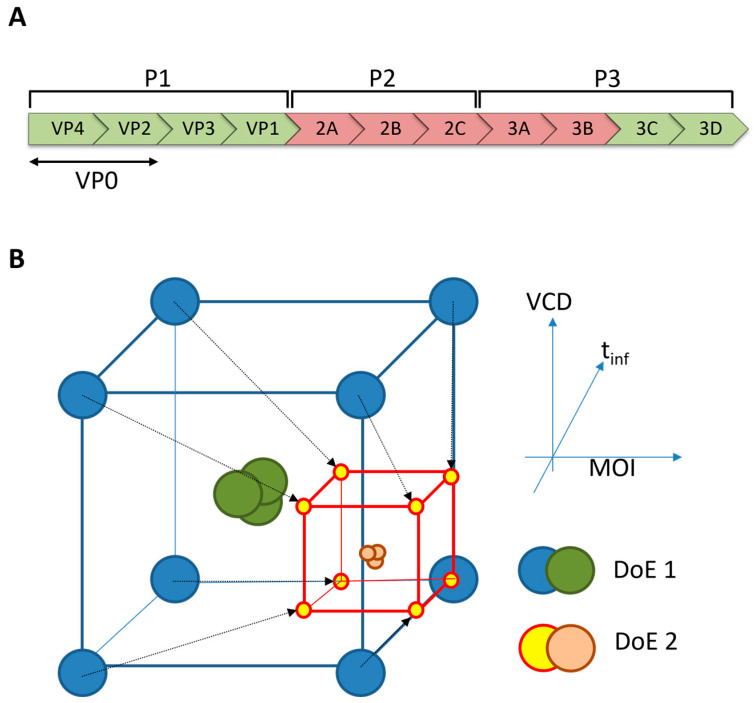
Enterovirus polyprotein and representation of the iterative process of Design of Experiments. (**A**) The enterovirus genome is composed of a single-stranded positive-sense RNA molecule that encodes a polyprotein of approximately 2200 amino acids. The polyprotein is subsequently cleaved by viral proteases into 11 mature proteins. Of these proteins, the P1 region encodes the structural proteins of the virion, while the P2 and P3 regions encode non-structural proteins that are involved in viral replication and assembly. Previous studies have shown that the expression of only the P1 capsid proteins and the 3CD protease (highlighted in green) is sufficient to produce enterovirus-like particles [14,15,16,17,18,19]. (**B**) The input factors for the DoE (represented along the X, Y, and Z-axes): multiplicity of infection (MOI), viable cell density at the time of infection (VCD), and infection period (t_inf_), respectively. These input factors are the culturing conditions that are subjected to optimization to maximize the output (viral protein production). The blue and green spheres represent the boundary conditions and triplicate experiments of the first iteration of DoE, respectively. Once optimal conditions are determined from the first iteration, the input factors are shifted towards the newly set boundary conditions of the second iteration, represented by the yellow/red and orange spheres. The process continues until the desired optimal conditions are reached.

**Figure 2 pathogens-14-00118-f002:**
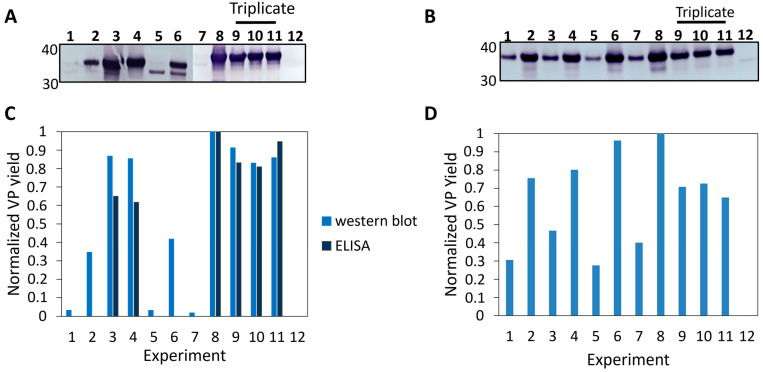
Analysis of the two iterations of the DoE for CVA6. (**A**) First iteration of DoE; western blot against CVA6 VP1. (**B**) Second iteration of DoE; western blot against CVA6 VP1. (**C**) Normalization of CVA6 western blot band intensities of the first iteration of the DoE using GelAnalyzer software v23.1 (light blue). Normalized ELISA against CVA6 A-particle (dark blue). The triplicates indicated a low biological variance, and the most promising fractions were found to be 3, 4, 6, and 8–11. (**D**) Normalization of CVA6 western blot band intensities of the second iteration of the DoE using GelAnalyzer software. Experiments 9–11 are triplicates indicating highly similar bands.

**Figure 3 pathogens-14-00118-f003:**
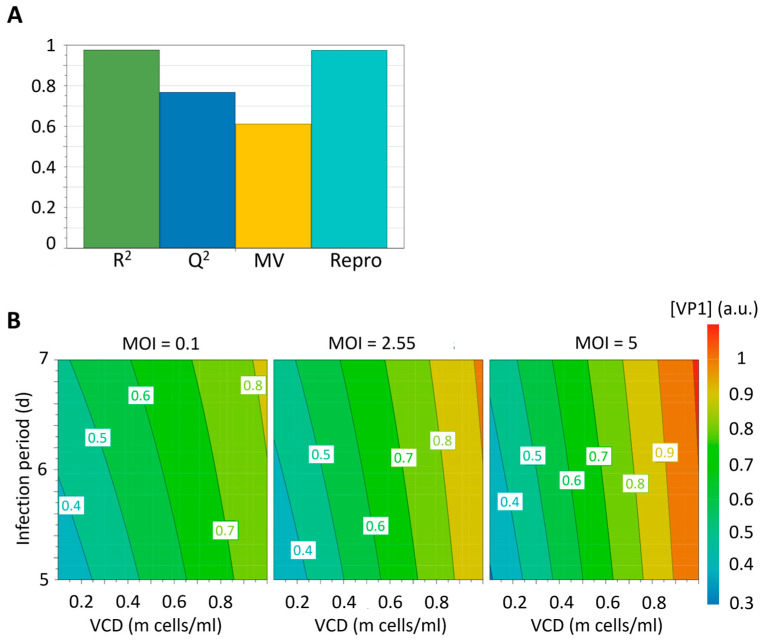
Summary of Fit and contour plots. (**A**) Summary of the fit plot for the second iteration of DoE for CVA6. R^2^ is an indicator for goodness of fit. Q^2^ estimates the prediction power of the model. MV is the model validity and measures the robustness of the model. Repro. is the reproducibility and provides a measure of the variability of the replicates relative to the variability of the dataset. Performance parameters of the model for CVA6: R^2^ = 0.976 (>0.5) indicated significance. Q^2^ = 0.768 (>0.5) determined a good model. MV = 0.65 (>0.5) indicated an absence in Lack of Fit, outliers, and transformation problems. Repro. = 0.97 (>0.5) indicated marginal biological variance. (**B**) Contour plots for CVA6 at the three different measured MOIs.

**Figure 4 pathogens-14-00118-f004:**
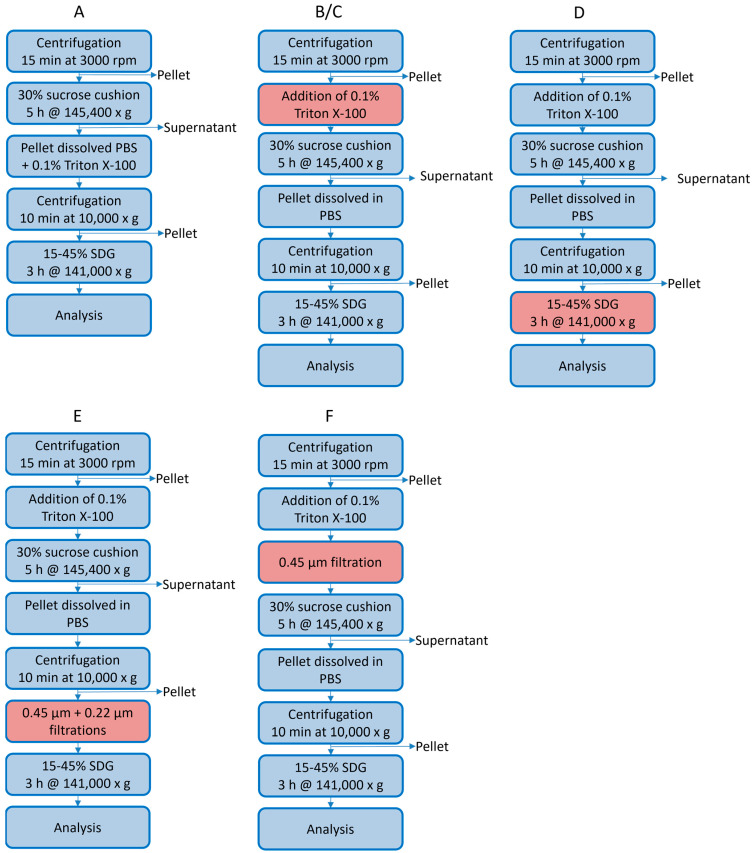
Overview investigated EV71 VLP DSP strategies. Strategy A was obtained from literature [16], and the most successful protocol (strategy F) was obtained after stepwise optimization (A through F). Red boxes indicated changes from the preceding DSP strategy. (**A**) Strategy for the DSP of EV71 VLPs produced at a small scale (50 mL) from the extracellular environment protocol was adapted from [16]. (**B**) As in Strategy A, but with Triton X-100 treatment prior to the sucrose cushion. (**C**) As in strategy B with a scaled-up culture volume (400 mL). (**D**) As in Strategy C, but with an increased volume (from 10 mL to 30 mL final volume) of the sucrose density gradient. (**E**) As in Strategy D, but with additional clarification (filtration) steps prior to SDG. (**F**) As in strategy E, with additional clarification (filtration) step prior to the sucrose cushion. This was the most successful protocol and maintained the standard for our EV71 VLP purifications.

**Figure 5 pathogens-14-00118-f005:**
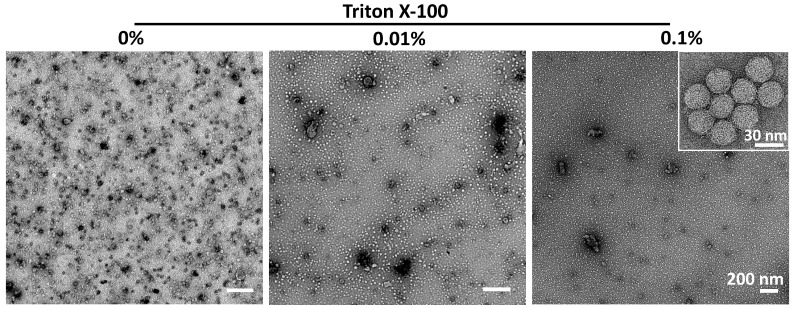
Investigation of the effects of low concentration of Triton X−100 on extracellular vesicles and VLPs. Since the majority of the extracellular vesicles (EVs) originate from membrane shedding, supplementation of detergent should break the polar and hydrophobic interactions. Triton X-100 was used to investigate its effects on extracellular vesicles and VLPs (scale bar = 200 nm). Inset of the image indicates the detectable presence of VLPs (scale bar insets = 30 nm). A Triton X-100 concentration below 1% does not inhibit viral protein assembly into VLPs, while significantly reducing or eliminating extracellular vesicles.

**Figure 6 pathogens-14-00118-f006:**
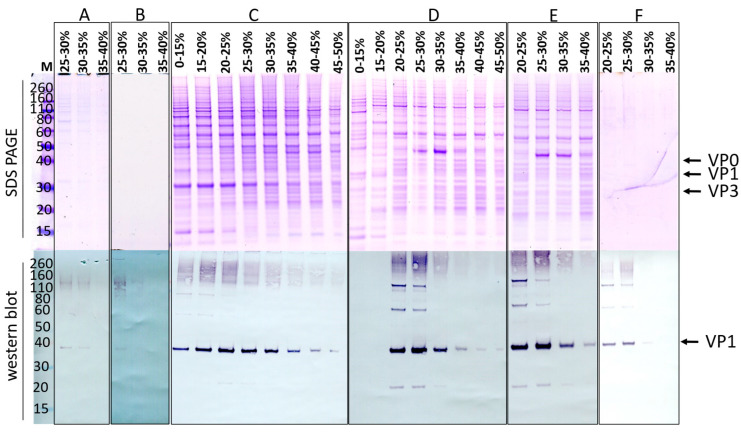
SDS PAGE gels and Western blots for all VP1-indicating samples of the SDGs of each DSP strategy. All DSP strategies (**A**–**F**) are described in Figure 4. Running conditions are described in Section 2.3 SDS PAGE & Western blot. The proteins found on SDS PAGE are hypothesized to be viral proteins VP0 (38 kDa), VP1 (35 kDa), and VP3 (27 kDa).

**Figure 7 pathogens-14-00118-f007:**
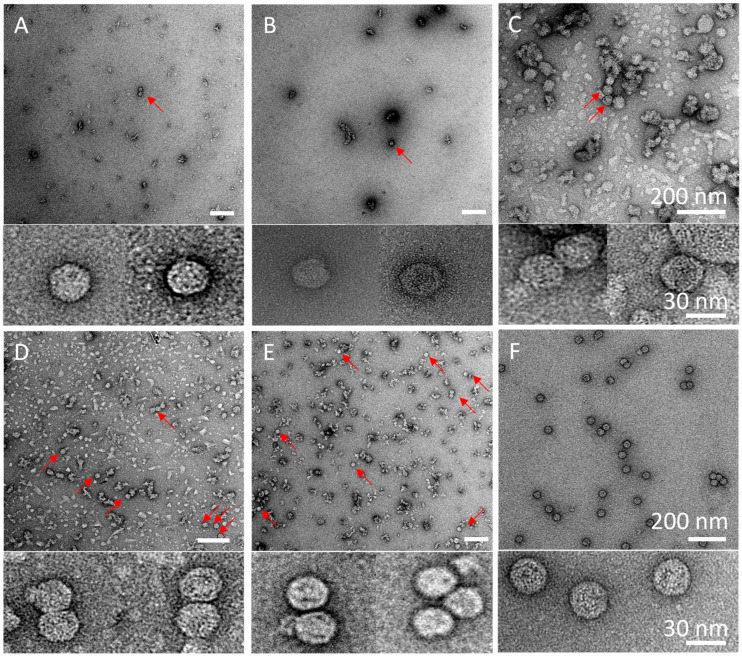
Electron microscopy images of the most promising fractions of the SDGs for strategies (**A**–**F**). All DSP strategies (**A**–**F**) are described in Figure 4. Experimental conditions are described in Section 2. White scale bars in the labeled images represent 200 nm, whereas in the insets, the scale bars represent 30 nm. For each strategy, two examples of VLPs are presented below the images.

**Figure 8 pathogens-14-00118-f008:**
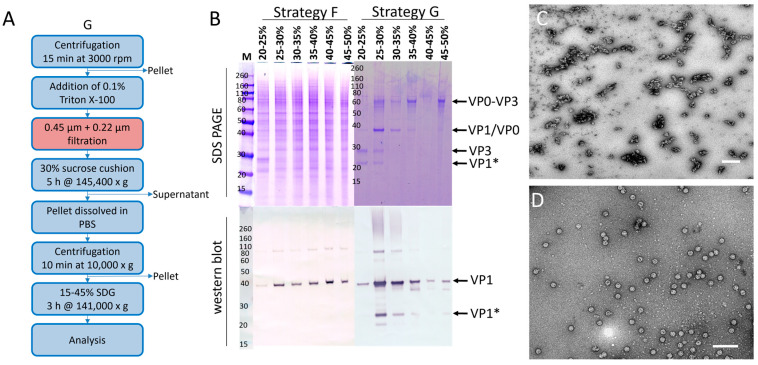
CVA6 VLP purification process. (**A**) CVA6-specific DSP strategy for the purification of 400 mL cell cultures containing VLPs. (**B**) SDS PAGE gels and western blots for all VP1-indicating samples of the SDGs of both CVA6 VLP DSP strategies. *: A truncated version of VP1. (**C**) Electron microscopy image of the most promising fraction of the SDG obtained by employing strategy F on CVA6 VLPs. The white scale bar indicates 200 nm. (**D**) Electron microscopy image of the most promising fraction of the SDG obtained by employing strategy G on CVA6 VLPs. The white scale bar indicates 200 nm.

**Table 1 pathogens-14-00118-t001:** Infection conditions and design matrix first iteration of design of experiments for CVA6. Cultures were infected using the infection parameters presented here. All 12 cultures were derived from the same preculture and diluted accordingly. The design matrix indicates the experiment based on the design space (Figure 1B), where −1 and 1 represent the lower and upper boundaries of the specific infection parameter. Experiments 9–11 are considered triplicates as they have the same infection conditions as the center of the design space (Figure 1B). Experiment 12 was the negative control (mock infection) that did not receive any virus.

Exp No.	Multiplicity of Infection	Infection Period (d)	Cell Density (Cells/mL)	Design Matrix		
				MOI	Infection Period	Cell Density
1	0.01	2	5 × 10^5^	−1	−1	−1
2	5	2	5 × 10^5^	1	−1	−1
3	0.01	7	5 × 10^5^	−1	1	−1
4	5	7	5 × 10^5^	1	1	−1
5	0.01	2	2 × 10^6^	−1	−1	1
6	5	2	2 × 10^6^	1	−1	1
7	0.01	7	2 × 10^6^	−1	1	1
8	5	7	2 × 10^6^	1	1	1
9	2.505	4.5	1.25 × 10^6^	0	0	0
10	2.505	4.5	1.25 × 10^6^	0	0	0
11	2.505	4.5	1.25 × 10^6^	0	0	0
12	n/a	7	1.25 × 10^6^	NC	NC	NC

**Table 2 pathogens-14-00118-t002:** Infection conditions and design matrix second iteration of design of experiments for CVA6. Cultures were infected using the infection parameters presented here. All 12 cultures were derived from the same preculture and diluted accordingly. The design matrix indicates the experiment based on the design space (Figure 1B), where −1 and 1 represent the lower and upper boundaries of the specific infection parameter. Experiments 9–11 are considered triplicates as they have the same infection conditions as the center of the design space (Figure 1B). Experiment 12 was the negative control (mock infection) that did not receive any virus.

Exp No.	Multiplicity of Infection	Infection Period (d)	Cell Density (Cells/mL)	Design Matrix		
				MOI	Infection Period	Cell Density
1	0.1	5	1 × 10^5^	−1	−1	−1
2	0.1	5	1 × 10^6^	−1	−1	1
3	0.1	7	1 × 10^5^	−1	1	−1
4	0.1	7	1 × 10^6^	−1	1	1
5	5	5	1 × 10^5^	1	−1	−1
6	5	5	1 × 10^6^	1	−1	1
7	5	7	1 × 10^5^	1	1	−1
8	5	7	1 × 10^6^	1	1	1
9	2.55	6	5.5 × 10^5^	0	0	0
10	2.55	6	5.5 × 10^5^	0	0	0
11	2.55	6	5.5 × 10^5^	0	0	0
12	n/a	7	5.5 × 10^5^	NC	NC	NC

## Data Availability

Full gels and Western blots can be found in the Appendix A.

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
