# Peer review of "Optimization of Enterovirus-like Particle Production and Purification Using Design of Experiments"

_pathogens, 2025, doi:10.3390/pathogens14020118_

Round 1

Reviewer 1 Report

Comments and Suggestions for Authors

Kuijpers et al., in “ Optimization of Enterovirus-like particle production and purification using the DoE," reported the production and purification process of CVA6 and EV71 VLPs using the BacV system. The DoE system effectively determined the optimal experimental conditions for producing VLPs. Validation of produced VLPs by WB, ELISA, and TEM was a valid methodology. Similarly, the different approaches for DSP for EV71 and CVA6 VLPs were also valid in vaccine production. The manuscript addresses the wet lab approach to effectively producing the VLPs.

Here are some minor corrections:

1.      Line #207: add references for the extreme conditions from the literature.

2.      Line#221: height can be replaced with molecular weight.

3.      Figure 2A: Is there any explanation for the lower MW band at lane 6? Particularly, does it correlate with the lane 5 band at the same MW?

4.      Figure 2A looks like lanes 1-6 and 7-12 ran separately on a different gel, unlike Figure 2B, western blot.

5.      The bar graphs can also have their own identity, like 2C and 2D, which will help the reader.

6.      The reviewer wondered why the ELISA was not performed in the 2nd iteration.

7.      The reviewer assumes that lane eight from Figure 2A and B were processed for further purification.  

8.      If possible, reduce the acronym usage. Some acronyms appeared less than a handful of times, confusing readers, particularly EVs.

Author Response

See Word document attached

Reviewer 2 Report

Comments and Suggestions for Authors

Summary

This manuscript is to optimize the production and purification procedure of coxsackievirus A6 (CVA6) and enterovirus A71 (EV71) virus-like particles. Using Design of Experiment, the authors optimize multiplicity of infection (MOI), viable cell density (VCD) and the infection time to reach a highest VLP production. The downstream purification processes are also optimized to obtain VLPs with high purity. The manuscript is well organized with clean data and will provide insights to development of the commercial enterovirus VLP vaccines.

1.        In Materials and Methods, “Viral protein production & purification” part, the authors have described a very detailed procedure. But more information should be clarified. Which ultracentrifugation tube is used? Which ultracentrifuge rotor is used? What volume of sucrose cushion or gradient is added? The volume of supernatant subjected to ultracentrifugation? The ultracentrifugation temperature?

2.        The medium MOI is 2.505 in the first iteration, but 2.55 in the second iteration. Is that a mistake or deliberate design? The authors are suggested to clarify the adjustment. 

3.        In Fig. 2A, the authors normalize the CVA6 VP1 band intensities to the highest output. However, the blot seems to derive from 2 individual film. It’s not appropriate to normalize the band intensities from different films, since the band intensity in the western blot can be affected by various factors, including exposure time, antibody freshness and concentration. I would suggest the authors to rerun the samples in Fig. 2A in a same gel as Fig. 2B if the authors have extra samples. 

Author Response

See Word document attached
